# WA-ResUNet: A Focused Tail Class MRI Medical Image Segmentation Algorithm

**DOI:** 10.3390/bioengineering10080945

**Published:** 2023-08-08

**Authors:** Haixia Pan, Bo Gao, Wenpei Bai, Bin Li, Yanan Li, Meng Zhang, Hongqiang Wang, Xiaoran Zhao, Minghuang Chen, Cong Yin, Weiya Kong

**Affiliations:** 1College of Software, Beihang University, Beijing 100191, China; 2Department of Obstetrics and Gynecology, Beijing Shijitan Hospital, Capital Medical University, Beijing 100038, China; 3Department of MRI, Beijing Shijitan Hospital, Capital Medical University/Ninth Clinical Medical College, Peking University, Beijing 100038, China

**Keywords:** long-tailed distribution, medical image segmentation, attention mechanism, class rebalancing

## Abstract

Medical image segmentation can effectively identify lesions in medicine, but some small and rare lesions cannot be well identified. Existing studies do not take into account the uncertainty of the occurrence of diseased tissue, and the problem of long-tailed distribution of medical data. Meanwhile, the grayscale image obtained from Magnetic Resonance Imaging (MRI) detection has problems, such as the features being difficult to extract and invalid features being difficult to distinguish. In order to solve these problems, we propose a new weighted attention ResUNet (WA-ResUNet) and a class weight formula based on the number of images contained in the class, which improves the performance of the model in the low-frequency class and the overall effect of the model by improving the degree of attention paid to the valid features and invalid ones and rebalancing the learning efficiency among the classes. We evaluated our method on an uterine MRI dataset and compared it with the ResUNet. WA-ResUNet increased Intersection over Union (IoU) in the low-frequency class (Nabothian cysts) by 21.87%, and the overall mIoU increased by more than 6.5%.

## 1. Introduction

A nabothian cyst is a common gynecological pathology in women of childbearing age, small in size and multiple in occurrence. Although, in most cases, this disease does not require targeted treatment, in some cases they can grow large [1], and multiple large nabothian cysts located in the interstitium of the cervix can lead to significant enlargement of the cervix [2]. Cervical hypertrophy and squamous metaplasia may trigger persistent chlamydial infection followed by life-threatening diseases, such as Human Papilloma Virus (HPV) [3].

Previously, ultrasonography was the first choice for imaging gynecological diseases because of its ease of operation, rapid diagnosis, and cost advantages, but there are some errors in the localization and counting of small soft tissue structures, such as small cysts (e.g., <0.5 cm in diameter). In contrast, Magnetic Resonance Imaging (MRI) provides good soft tissue contrast and has multiple reconstructed planes and a large field of view to better describe tissue structure [4]. The specific characteristics of this technique include its multiplanar imaging capability, tissue specificity, and the ability to distinguish layers within the uterine wall. Additionally, it can differentiate adnexal tissue, uterine tissue, and pelvic fat; thus, it is highly beneficial for imaging gynecological diseases [5]. Magnetic resonance imaging can accurately detect the location, size, number and other information concerning nabothian cysts. MRI examination is free of X-ray radiation, which means it does not cause any damage to the human body. Meanwhile, it has high tissue resolution, which provides clearer and richer imaging information and is beneficial for female reproductive system examination and diagnosis, and, as a result, it is increasingly used clinically to examine lesions within the uterus.

Medical image segmentation usually involves segmenting the tissues or organs of the diseased part. Manual segmentation is a difficult task as doctors and experts need to draw the boundaries directly in medical images. This method depends on the experience and knowledge of the segmenters and is highly subjective, as well as very inefficient. The importance lies in automatic segmentation of uterine MRI images through artificial intelligence and deep learning.

U-Net and its variants are the main models used and studied in medical image segmentation at present. Ronneberger et al. [6] proposed the U-Net network structure based on the Fully Connected Network (FCN) [7,8] in 2015, which is an encoder–decoder structured network that has an excellent performance in medical image segmentation. In 2018, the ResUNet model [9] utilized Res-block, inspired by ResNet [10], instead of normal convolution. In addition, Gu et al. [11] proposed Ce-net to retain more spatial information. Zhou et al. [12] proposed Unet++, which enhances the acquisition of features at different levels, mainly through densely nested skip links. Jha et al. [13] proposed an automated pixel-level segmentation model Resnet++ based on Unet++. In 2020, Ibtehaz et al. [14] proposed the MutiRes module that introduced the residual path ResPath instead of the skip connection. Lou et al. [15] replaced the encoder and decoder with a Convolutional Neural Network (CNN) architecture and used a residual module instead of skip connections in 2021. However, these models are not friendly to the low-frequency class and pay insufficient attention to it, which tends to ignore some features, resulting in poor effects in general.

Attention mechanisms are a common way to increase the focus of models on major features. In 2014, Mnih et al. [16] used the attention mechanism for the first time on a Recurrent Neural Networks (RNN) model for image classification. In 2015, Xu et al. [17] proposed soft attention (standard backpropagation technique) and hard attention (maximizing variable lower bound). In 2017, Woo et al. [18] proposed the Cbam module that combines Spatial Attention and Channel Attention mechanisms. Fu et al. [19], in 2019, introduced a dual attention channel mechanism in the segmentation task to enhance features in the spatial dimension and channel dimension, respectively. In 2021, Dai et al. [20] proposed a multiscale channel attention module to better fuse semantic and scale-inconsistent features. Hou et al. [21] embedded location information into channel attention to simultaneously capture remote dependencies and maintain accurate location information through two one-dimensional features. Liu et al. [22] maintained high internal by polarization filtering in both channel and spatial attention computation resolutions. Yang et al. [23] proposed optimizing the energy function to find the importance of each neuron based on some well-known neuroscience theories. Liu et al. [24] proposed a regularized channel attention module (NAM). Recently, Ruan et al. [25] proposed MALUnet, a lightweight network containing four modules, DGA, IEA, CAB and SAB.

Long-tailed distribution is a common but difficult problem. Most datasets in medicine suffer from severe class inhomogeneity [26]. This is a classification situation in which the distribution of data or labels is uneven and unbalanced. In multi-class segmentation, this is reflected by the fact that high-frequency classes are not even within the same data magnitude as low-frequency classes, in terms of both the number of pixel points and the number of images included in the training dataset. The main methods to solve the long-tailed distribution problem mainly include feature manipulation methods [27,28,29] and re-sampling methods [30]. Re-sampling methods are divided into over-sampling [31,32] and under-sampling [33,34]. However, their scope of use is limited and increases the training cost and difficulty of the model. Another way to solve this long-tailed problem is class rebalancing [35,36], but the vast majority of current work on long-tailed distributions has been directed at image classification and instance segmentation [37]. In semantic segmentation, a well-known method is based on the balance method in softmax strategy [38], which is to rebalance pixel classification based on the number of pixels the class contains. In long-tailed image classification, rebalancing can have an effect through the class frequency since the image samples are independently and identically distributed; however, in the semantic segmentation domain, the correlation of neighboring pixels leads to pixel samples that do not conform to this feature [37]. As a result, the strategy is prone to fail and negatively affect the Intersection over Union (IoU). The class weight formula, based on the number of pixel points, may find it challenging to improve the model inference effect.

The uterine multi-class segmentation task aims to identify common organ structures and diseased tissues inside the uterus, such as the outer uterine wall, uterine cavity, myomas of the uterus, and nabothian cysts, in MRI images of uterus. It is a challenging task, due to the large individual variability, high accuracy requirements, strong interferences between tissues and organs, uneven distribution of datasets, and difficulty in extracting gray-scale map features. To solve the above problems, we propose a new end-to-end network model, called Weighted Attention ResUNet (WA-ResUNet), which focuses more on low-frequency classes and can better extract positive features and effective features of low-frequency classes. We also propose a new class weight formula for loss function, based on the number of images the class includes in the weight formula, which enables an increase in IoU, while better balancing high-frequency classes and low-frequency classes, significantly enhancing low-frequency class IoU with a slight loss of high-frequency class IoU, resulting in considerable improvement in the mIoU of the model.

In summary, our efforts have yielded the following results:We proposed a novel network model WA-ResUNet. The model demonstrates the capability to significantly improve the performance of low-frequency classes.We proposed a novel class weight formulation specifically designed for multi-class semantic segmentation. It prioritizes the performance of IoU in the low-frequency class.We conducted extensive experiments on the uterine MRI dataset. The results review indicates the superiority of our model compared to the current commonly used advanced U-Net family of models and attention mechanisms.

## 2. Methods

In this section, we describe, in detail, the proposed WA-ResUNet (Weighted Attention Res-Unet) for processing semantic segmentation models of uterine MRI images and a new adaptive class weight formulation.

### 2.1. The Network Architecture of WA-ResUNet

The network structure of WA-ResUNet is shown in Figure 1. Based on the U-Net [6] model and ResUNet [39] model with a residual block, having a backbone network which is ResNet50 [10], we propose a channel residual block containing a channel attention mechanism to replace the original encoder structure used by U-Net, which is called the CA Res-Block (Channel Attention Res-block). Meanwhile, we use a spatial attention mechanism to design a new connection bridge, in order to improve U-Net’s skip connection structure, which we call SAB (Spatial Attention Bridge). In addition, we add an adaptive class weight formula to the calculation of the loss function, so as to balance the learning degree of high-frequency classes and low-frequency classes.

When inputting the image to be processed into the network, it undergoes an initial processing step where it is passed into the weight calculation part to determine the weight of its classes. Then a 3 × 3 convolution kernel is used for the convolution operation, and then it is passed through the maximum pooling layer, and input into four consecutive layers of CA Res-blocks. The connection between each two layers through downsampling enables it to obtain deeper information. Subsequently, the feature map output of the fourth layer residual block is input to the decoder stage. The input of each layer of the decoder consists of the output of the upper layer after upsampling and the output of each layer encoder through the SA Bridge after concat connection, and the output of the last layer is the final output.

The internal structure, basic principles and related functions in the network structure are introduced as follows.

### 2.2. Channel Attention Mechanism

In the field of medical images, identifying less significant features is usually a major difficulty in model learning. The U-Net network structure itself is shallow, which means its ability to acquire features is limited. Moreover, MRI images are grayscale images with blurred boundaries, noise, and poor contrast. This means it is difficult for traditional feature representation methods to capture the subtle differences in their features. At the same time, there are certain differences in the images of the same tissue for different patients, different modalities, and different imaging devices, and there may even be differences between different frames of the same modality [40]. The model has a high level of difficulty in learning non-apparent features.

Furthermore, since low-frequency classes are prevalent in the dataset, its learning ability in regard to insignificant features in the low-frequency classes becomes even less easy. As there are fewer discriminant bases for low-frequency classes, the model learning bias is larger, and some invalid features are more likely to be incorrectly learned, which greatly reduces the IoU of reasoning for low-frequency classes.

Therefore, we propose a new modular CA Res-Block based on the residual block to solve these problems. We designed this block based on the residuals of resnet50 in ResUNet. The residual block can help the model to extract more effective features more accurately. At the same time, inspired by Liu et al. [24], we applied a normalized channel attention mechanism to the field of multi-class medical image segmentation, which enables the model to better suppress features that have little impact on the model effect, and invalid features, by imposing a weight sparsity penalty on the model. In our CA Res-Block, the attention mechanism is inserted into the residual block after the third convolution, which allows it to combine the benefits of both methods. The CA Res-Block can effectively suppress the attention of the model to invalid features, which means that some interfering features can be removed in advance during the model learning process, so as to pay more attention to useful features to obtain more positive features more efficiently, and, thus, improve the ability of the model to recognize negative features and learn effective features, especially those in the low-frequency class.

Figure 2 shows the structure of the channel attention module incorporated in the CA Res-Block. The input feature maps are normalized through the Batch Normalization layer and then multiplied with the channel weights, which are the scale factors of the different channels, and are then output after activation by the sigmoid activation function combined with the input feature information. Its corresponding mathematical formula is expressed as Formulas (Equation 1) and (Equation 2).
(1)$$F_{out}=sigmoid\left(W_{\alpha }\times \left(\alpha \times \frac{F_{in}-\mu_{\zeta }}{\sqrt{\sigma_{\zeta }^{2}+\varepsilon }}+\beta \right)\right)$$
(2)$$W_{\alpha }=\frac{\alpha_{i}}{\sum_{k=0}^{n}\alpha_{k}}$$
where $F_{in}$ and $F_{out}$ represent input and output characteristics, respectively, $\mu_{\zeta }$ and $\sigma_{\zeta }$ are the mean and standard deviation of small batch ζ, respectively, β is the hyperparameter of displacement, α is the channel scale factor, and $W_{\alpha }$ is the weight of the corresponding channel α. The calculation process is shown in Formula (Equation 2).

### 2.3. Spatial Attention Mechanism

The advantage of the U-Net network is the skip connection. It combines depth, semantics and coarse-grained feature mapping from the decoder subnetwork with shallow, low-level and fine-grained feature mapping from the encoder subnetwork [12]. However, the combination process only fuses the features in the downsampling process into the corresponding hierarchy in the upsampling process through the method of concat. We believe that this method still has some flaws because it is likely to learn some irrelevant information at different levels. Moreover, in the process of combining features at deep and shallow levels, some insignificant features have to be ignored, due to the limited size of the feature map. Therefore, the direct combination of features through the traditional skip connection can easily lead to the removal of non-apparent positive features, while more invalid information is retained, thus affecting the final inference effect of the model. In addition, when the features of deep and shallow levels are combined, the location information of deep and shallow features in the feature map also deserves more attention. The local information of shallow features that more attention is directed at can be supplemented by deep features. If the network can pay more attention to important features, the local deep features can be combined in a more detailed and specific way during the supplement. Thus, fewer invalid deep features are transmitted upward, and the model’s ability to obtain effective information and inhibit invalid information is increased. Cbam [18] proposed that spatial attention mechanisms can play an important role in suppressing irrelevant information. At the same time, they can increase the model’s attention to spatial information. It is expected that better inference results can be obtained if the combination process allows the agent to pay more attention to the key areas and suppress irrelevant information.

Therefore, we propose a connection bridge, based on the spatial attention mechanism, which we call SA Bridge. In the connection process of SA Bridge, we input the upper-level features into the spatial attention module, filter out the invalid features, and then combine them with the feature map in the up-sampling process in a concat way, which means that, after the results of downsampling are achieved, through the spatial attention module, feature information with more effective position information is transferred to the upsampling process. At the same time, it can also enable the model pay more attention to the position information during the segmentation.

Figure 3 shows the spatial attention module structure we used. It first aggregates channel information from a feature map by applying average pooling and maximum pooling operations along the channel axis to generate two two-dimensional feature maps. Effective feature descriptors are generated by its concat connection, and then the two-dimensional spatial attention map is generated through the convolution layer. After applying the sigmoid activation function, this is reflected in the input feature map. The mathematical formula of this method is Formula (Equation 3).
(3)$$F_{out}=sigmoid\left(f^{7\times 7}\left(concat\left[AvgPool\left(F_{in}\right),MaxPool\left(F_{in}\right)\right]\right)\right)$$
where $F_{in}$ and $F_{out}$ represent input and output features, respectively, $f^{7\times 7}$ represents the convolution layer whose convolution kernel is 7×7, AvgPool and MaxPool represent the average and maximum pooling layers, respectively.

### 2.4. Adaptive Class Weighting Formula

There is a serious long-tailed distribution problem in our dataset. Table 1 shows the difference in the number of images and the number of pixels of the uterine wall, uterine cavity, myomas of the uterus and nabothian cysts in our dataset. The ratio between the number of images of the uterine wall and the nabothian cysts in the dataset is about 10:1. The ratio of the number of pixel points between the two classes is 450:1. In addition, the ratio of the number of pixel points between the uterine wall and the uterine cavity classes is more than 20:1. The obvious long-tailed distribution makes it difficult for the model to learn enough effective features for low-frequency classes.

The improved segmentation effect resulting from an increase in the number of pixel points is mainly attributed to the addition of new features that the network learns. However, in medical imaging, the increase in a single pixel does not necessarily introduce new features, since individual pixels lack labels, and the model cannot make accurate classifications based solely on one pixel. Additionally, the features brought by individual pixels are not easily learned in some cases. It is the added pixel points that constitute a completely new single instance of the class that is more acceptable to the model. Conversely, a small number of pixels are not directly related to a small number of features, because if the number of singleton occurrences is large enough, the network can also learn enough positive information with a small number of singleton pixels, because in this way, the common points between each singleton are easier to find.

In addition, in this dataset, due to the small number of pixels in singletons of the low-frequency class, the pixel-level weight formula leads to too low a weight in the model for high-frequency classes. This makes many features in the high-frequency classes easy to learn, having a great impact on the model’s effect in regard to that which is not learned, while the excessive weight of low-frequency classes leads to over-learning, wherein some invalid features are wrongly learned, leading to a significant decrease in model performance.

In the meantime, the class weight formula, based on the number of pixels, does not align with the assumption of mutual independence. As illustrated in Figure 4, using an example from our dataset, Table 2 displays the number of pixels in the example image. When employing a weight formula based on the number of pixels as, for instance, in the diagram, including a pixel belonging to the uterine wall also implies the presence of additional pixels (11,201 in this example). This is because a pixel within the uterine wall implies the existence of a larger connected region representing the uterus, which typically contains more than just one pixel. Consequently, the pixel-based class weight formula inadvertently increases the inverse proportional coefficient for classes with more pixels in the singleton, resulting in an excessive reduction of their weights.

In the domain of medical imaging, the presence of an augmented quantity of images encompassing classes signifies an escalation in the acquisition of novel instances. This consequentiality posits a higher likelihood of incorporating newly learnable features into the neural network. Furthermore, the assignment of weights, contingent upon the frequency of instances, is not entirely independent within the context of semantic segmentation. This is due to the potential relations that may exist between two instances of the same graph, such as their locations. Considering that our methodology relies on image samples, it adheres to the principles of an independent uniform distribution; thus, satisfying the requisites for class rebalancing.

Consequently, a novel adaptive formula for determining class weights is presented, specifically tailored for the task of multi-class semantic segmentation. This formula assigns additional weights to distinct classes, which is contingent upon the respective number of images representing each class within the training dataset. By virtue of this approach, the segmentation performance of the model is enhanced, particularly for those classes with low occurrence frequencies. This leads to an improvement in the mean Intersection over Union (mIoU) metric for such low-frequency classes. The weight allocation scheme for various categories, characterized by different image quantities, is depicted in Figure 5. As per the underlying principle, classes with higher occurrence frequencies are assigned relatively lower weights, while classes with lower occurrence frequencies are endowed with higher weights.

Firstly, the proportion coefficient of each class is calculated based on the inverse ratio of the number of pictures contained in each class. The formula is shown in Formula (Equation 4).
(4)$$P_{i}=\frac{\sum_{j=0}^{C}N_{j}}{N_{i}}$$
where $P_{i}$ represents the proportion of class *i* in every class, $N_{i}$ and $N_{j}$ represent the total number of pictures occupied by corresponding classes *i* and *j* in the dataset, and *C* represents the number of classes contained in the dataset.

Then, the proportion of every class is normalized and multiplied by a fixed coefficient to turn the weighted sum into the total number of classes. The mathematical formula expression is shown in Formula (Equation 5).
(5)$$W_{i}=C\times \frac{P_{i}}{\sum_{j=0}^{C}P_{j}}$$
where $W_{i}$ represents the weight assigned to corresponding class *i*, *C* represents the number of classes contained in the data set, and $P_{i}$ and $P_{j}$ represent the proportion of classes *i* and *j* in every class.

Therefore, combining these two formulae, our proposed class weight formula is shown in Formula (Equation 6).
(6)$$W_{i}=\frac{C}{N_{i}\times \sum_{j=0}^{C}\frac{1}{N_{j}}}$$
where $W_{i}$ represents the weight assigned to corresponding class *i*, *C* represents the number of classes contained in the data set, and $N_{i}$ and $N_{j}$ represent the total number of images occupied by corresponding classes *i* and *j* in the data set.

## 3. Results and Discussion

In this section, we undertake an evaluation of the segmentation efficacy offered by the WA-ResUNet model, while simultaneously analyzing the impact it exerts. Initially, as a comparative benchmark, we implemented both a U-Net model and a ResUNet model, devoid of any attention mechanisms, in order to demonstrate the progress attained by our proposed model. Ablation experiments were further conducted on both the model architecture and weight formula, and, subsequently, the outcomes were compared against other widely recognized advanced approaches. The obtained experimental results are meticulously analyzed and comprehensively discussed.

### 3.1. Datasets

In this section, we conducted extensive experiments on our uterine MRI dataset to train and evaluate our model.

Our uterine MRI dataset comprised a total of 3206 original grayscale MRI images, each accompanied by its corresponding annotated counterpart. The dataset encompassed four distinct classes, namely the uterine wall, uterine cavity, myomas of the uterus, and nabothian cysts. Each class was assigned a specific pixel value (1, 2, 3, 4) within the real-valued images. To establish a clear division, the dataset was partitioned into a training set and a validation set, with a ratio of approximately 6:1. The training set consisted of 2738 images, while the validation set comprised 468 images. It is important to note that certain images solely depicted the background and were, thus, removed prior to training. Consequently, the effective training set encompassed 1036 images. Figure 6 provides a partial sample showcasing some images from the dataset.

### 3.2. Ablation Experiment

To validate the efficacy resulting from the enhancements made to various components of our proposed model, ablation experiments were conducted on the WA-ResUNet model. Table 3 presents the comprehensive experimental results corresponding to the WA-ResUNet model. The Overall column represents the aggregate performance of the model across all classes. Each row in the table depicts the model’s inference outcome subsequent to the addition of a newly improved component, building upon the enhancements introduced in the preceding row.Consequently, the first line of the table represents the outcome achieved by employing the U-Net model independently. The second line portrays the results obtained through the incorporation of the Res-Block onto the U-Net model. Moving forward, the third line demonstrates the performance achieved by simultaneously applying Res-Block and the weight formula to the U-Net architecture. The fourth line signifies the incorporation of the CA Res-Block, weight formulae, and the channel attention mechanism onto the U-Net model. Finally, the fifth line encapsulates the comprehensive model, encompassing the utilization of the CA Res-Block, weight formulae, and the SA Bridge, in conjunction with the U-Net architecture.

The experimental results clearly indicate that each component of our proposed WA-ResUNet model contributes to the improvement of the Intersection over Union (IoU) for the low-frequency class (nabothian cysts), as well as the overall mean IoU (mIoU). This substantiates the efficacy of individual components within WA-ResUNet and further verifies the cumulative effect achieved through their integration. Specifically, the combination of these components leads to enhanced model inferences, thereby demonstrating the potential for further improvements in the reasoning capabilities of the model.

#### 3.2.1. Ablation Experiment of Network Structure

In this section, we conducted ablation experiments on several parts of the network structure to demonstrate the effectiveness of each of the modules we added, and their boosts when combined.

With our dataset, Table 4 shows the comparison of the IoU, sensitivity and precision results of the original ResUNet model without any attention module, the ResUNet model with the addition of the channel attention module, the ResUNet model with the addition of the spatial attention module, the ResUNet model with the addition of both the channel attention module and the spatial attention module for the segmentation with the weight of all classes being 1.

From the results obtained, it is evident that employing either of the two attention mechanisms without the use of the weight formula alone resulted in a noticeable enhancement in the network’s segmentation performance for low-frequency classes, as well as an improvement in the overall segmentation ability. Moreover, the simultaneous utilization of both attention mechanisms further amplified the network’s capacity to learn from low-frequency classes, leading to a more pronounced boost in the overall segmentation performance of the network.

Table 5 shows the comparison of the IoU, sensitivity and precision results of the original ResUNet model without any attention module, the ResUNet model with the addition of the channel attention module, the ResUNet model with the addition of the spatial attention module, the ResUNet model with the addition of both the channel attention module and the spatial attention module for the segmentation with the new weight formula.

The results clearly indicate that the channel attention module and the spatial attention module exhibited individual efficacy, as their standalone integration yielded improvements in terms of IoU, sensitivity, and precision across nearly all classes, as well as on the overall metrics of mIoU, sensitivity, and precision. Notably, while the combined utilization of both attention mechanisms slightly compromised the segmentation performance for the uterine wall and myomas of uterus, it significantly enhanced the segmentation outcomes for the lower frequency classes, namely the uterine cavity and nabothian cysts. Furthermore, the overall mIoU and sensitivity demonstrated substantial improvements as a result. Although there was a slight decrease in precision, it remained within acceptable limits.

To further demonstrate the advanced nature of our proposed network structure, we compared the current representative advanced models under the new weight formula, and they finally presented the segmented IoU results, as shown in Table 6.

Our model demonstrated a decline in performance for the high-frequency class. However, owing to its exceptional performance in the low-frequency class, WA-ResUNet outperformed the influential existing models in terms of mIoU. These findings indicate that our model exhibits a greater emphasis on low-frequency classes, rendering it more suitable for scenarios where such classes hold relatively high significance; although this particular focus may have adverse effects on high-frequency classes.

In addition, we also compared it with the more advanced attention mechanism modules currently available. To compare the attention module, we applied it to the U-Net model with ResNet50 as the backbone to eliminate the interference of other factors, and they finally presented the segmentation IoU results, as shown in Table 7.

In comparison to the prevailing and contemporary attention mechanisms, our model demonstrated a remarkable improvement, of over 9%, in IoU for low-frequency classes when compared to the existing attention mechanisms that exhibit the highest performance in this particular class. Additionally, our model surpassed the second-ranked approach by achieving a 1.8% enhancement in mIoU. These outcomes affirm the excellence and advances of our research findings in the domain of attention mechanisms. This indicates that our SA bridge and CA Res-Block are capable of acquiring more salient information pertaining to the low-frequency classes, although some valid features in high-frequency classes may be incorrectly removed in the process.

#### 3.2.2. Ablation Experiment on Weight Formula

In this part, we demonstrate the effectiveness of our proposed weight formula by comparing the performance of the ResUNet model and the WA-ResUNet under three different weight formulae.

In our dataset, IoU, the sensitivity and precision results of the ResUNet model and the WA-ResUNet under different weights are shown in Table 8. Weight 1 denotes an equal weight assignment of 1 in all classes. Weight 2 corresponds to the weight distribution determined by the class weight formula based on the number of pixels. Weight 3 signifies the weight allocation determined by our proposed weight formula.

In both ResUNet and WA-ResUNet architectures, the traditional formula exhibited a noticeable sensitivity advantage; however, this advantage was accompanied by a substantial decrease in precision, consequently resulting in lower IoU values. Notably, in ResUNet, the formula utilizing default weights performed considerably more poorly than the new weight formula for classes characterized by a long-tailed distribution. Ultimately the new weight formula yielded superior mIoU results. This advantage was further amplified when spatial and channel attention mechanisms were incorporated into the WA-ResUNet model, wherein the new weight formula proved to be more effective in handling low-frequency classes, manifesting remarkable improvements in IoU, sensitivity, and precision. Additionally, the new weight formula exhibited a significant advantage in total precision, surpassing the default formula by more than 6%.

To further demonstrate the universality of our proposed weight formula, we applied the new weight formula to some representative models, and their IoU experimental results under the default weights and the new weight formula were compared, as shown in Table 9. The default weight was 1 for all classes.

Based on the experimental findings, it is evident that our proposed weight formula yielded improved performance in comparison to several conventional models. In most scenarios, the models exhibited significantly enhanced ability to learn from classes characterized as nabothian cysts and uterine cavity, without incurring excessive losses in high-frequency classes, such as the uterine wall and myomas of the uterus. This allows the models to achieve overall improvements in performance in terms of mIoU. Furthermore, our weight formula also demonstrated a positive impact on other classes in certain models, over and above the uterine cavity and nabothian cysts. We attribute this observation to the formula’s ability to reduce the model’s attention to invalid features in high-frequency classes to a certain extent. However, it should be noted that the results obtained lack sufficient stability as, in certain instances, they led to incomplete learning of valid features. Moving forward, our future research will prioritize the exploration of effective techniques for more precise elimination of irrelevant features.

In our experimental study utilizing the uterine MRI dataset, among the four classes considered, the lesion class holds significantly higher importance compared to the organ class. This distinction arises due to the primary objective of the task, which aims to facilitate accurate diagnosis by medical practitioners. Furthermore, our proposed modules exhibit versatility in accommodating multiple classes within various networks. For instance, the integration of the weight formula module in several other models, including DC-UNet and Ce-net, leads to a simultaneous increase in IoU for two focal classes. Therefore, we consider that our modules and model are well suited to the task.

### 3.3. Comparative Experiments

In this part, we conducted sufficient comparative experiments between WA-ResUNet and other models to show the progress of our model and reflect the advanced nature of WA-ResUNet.

We first compared the WA-ResUNet model under the new weights with the traditional U-Net model and the ResUNet model under the default weights to show the baseline improvement. Table 10 compares IoU, sensitivity and precision under different classes of the U-Net model, ResUNet model and WA-ResUNet, as well as the mIoU of the whole class in our dataset.

In comparison to ResUNet, the WA-ResUNet model exhibited a slight decrease in IoU specifically related to the uterine wall class. However, it demonstrated significant improvements in the remaining three classes, particularly in the low-frequency class of nabothian cysts, where WA-ResUNet achieved a noteworthy enhancement of 21.87%. Moreover, the WA-ResUNet model showcased substantial advancements in sensitivity and precision for the uterine cavity and nabothian cysts. Specifically, in the uterine cavity, sensitivity increased by 2% and precision by over 4%. Notably, there was a remarkable increase of sensitivity by 1 times (16.64%) and precision by 6 times (61.03%) for nabothian cysts. Overall, the precision of the WA-ResUNet model experienced an impressive increase of over 15%. Furthermore, there was a notable improvement of 6.5% in mIoU and more than 5% in average sensitivity.

In comparison to U-Net, the enhanced WA-ResUNet model showcased improvements across all IoU classes, with a substantial enhancement of nearly 20% observed in the lowest-frequency class and over 10% in the second lowest-frequency class. Notably, the sensitivity performance was significantly improved across all classes, particularly noticeable in an impressive improvement of over 15% achieved in the lowest-frequency class and more than 12% in the second lowest-frequency class. Regarding precision, the WA-ResUNet experienced a slight decline in performance specifically related to the uterine wall class, but outperformed U-Net in the remaining classes, achieving a remarkable improvement of nearly 55% in regard to nabothian cysts. Overall, the WA-ResUNet model exhibited significant improvements compared to U-Net, with an increase of more than 9% in mIoU, approximately 9% in sensitivity, and 14% in precision.

In addition, we compared it with some of the other models that are currently more advanced. Table 11 shows the experimental results of comparisons between the WA-ResUNet model and other models IoU and frames per second (FPS).

It is evident that while UNet++ exhibited superior performance in the three high-frequency classes, namely the uterine wall and myomas of the uterus, intensive feature sampling caused the network to extract more invalid features incorrectly and reduced the model’s attention to the positive features in the low-frequency class. Consequently, it performed poorly on low-frequency class nabothian cysts and rarely split singletons correctly. As a result, UNet++ achieved an overall mIoU of only 46.11%. Furthermore, the performance of other models in the low-frequency class significantly lagged behind that of the WA-ResUNet. In comparison, the effect of the WA-ResUNet on nabothian cysts exhibited an improvement of over 14% when compared to the second-best performing model, Channel-UNet. Additionally, the WA-ResUNet attained an mIoU exceeding 51%, on average, while other models fell below the 50% mark. Furthermore, barring Ce-net and Resunet++, which demonstrated evident inefficiency, our model’s inference speed remained comparable to, and sometimes even surpassed, that of many other models. This result highlights the capacity of our model to deliver improved inference outcomes without compromising speed. Although our method did not achieve the optimal frames per second (FPS), it only required 4 s to analyze a patient (based on 24 images per patient). This speed was 360 times faster than the manual analysis time frame, which typically necessitates two days to complete the process and yield results.

### 3.4. Visual Experimental Effect

To provide a clearer visualization of the improvement in the segmentation effect of the low-frequency class achieved by our model, we conducted visualization processing on the experimental reasoning results. The visualized results are presented in Figure 7 and Figure 8.

Figure 7 mainly reflects the overall predicted situation and describes the difference between the precision and sensitivity ratio of the predicted results, and Figure 8 shows the improvement of the model in the low-frequency class (nabothian cyst).

### 3.5. Publish Data Set Experiment Results

In order to prove the universality of the model and method proposed in this paper, the WA-ResUNet was applied to a publicly available MRI dataset of abdominal organs [42], which included four segmentation classes: liver, left kidney, right kidney and spleen. The corresponding number of pixels and the number of images included are shown in Table 12.

From an examination of Table 12, it is evident that this dataset exhibited a pronounced long-tailed distribution issue. Additionally, the MRI image data in this dataset conformed to the training and inference characteristics of the model. Regarding the abdominal organ MRI dataset, Table 13 presents the segmentation results of the U-Net model and the WA-ResUNet model. It is worth mentioning that the liver class contained a greater number of images compared to the other three classes. Furthermore, in terms of the number of pixels, there was a noticeable disparity between the proportion of liver pixels and the pixels belonging to the other three classes.

The data presented in Table 13 reveal that the enhanced WA-ResUNet model yielded significant improvements in the intersection ratio between the left and right kidneys in the low-frequency class, when compared to the traditional U-Net model. Specifically, the WA-ResUNet model demonstrated an approximate 12% increase in the intersection ratio for the left kidney, albeit demonstrating a slight decrease for the liver and spleen in the high-frequency class. However, this negligible decrease in the high-frequency classes contributed to an overall improvement of 2.80% in mIoU. Additionally, the improved WA-ResUNet model exhibited notable enhancement in sensitivity, particularly in the low-frequency class and in overall performance. The sensitivity for the left kidney demonstrated an improvement of approximately 13%. In terms of accuracy, the WA-ResUNet model displayed significant improvements across all classes compared to U-Net. Notably, the accuracy for the liver, left kidney, right kidney, and spleen improved by 5%, 0.01%, 5%, and 1%, respectively. Overall, the WA-ResUNet model achieved an improvement of over 2% in accuracy.

We advocate that the WA-ResUNet model possesses the capability to mitigate the deleterious effects of long-tailed distribution. By enhancing the learning rate of the network in extracting meaningful features, and mitigating the negative impact of irrelevant features, the network’s capacity to learn from low-frequency classes is enhanced. Consequently, the network significantly improves in performance for low-frequency classes, even though there may be a slight decrease in performance for high-frequency classes. This phenomenon can be attributed to the decreased weight assigned to the high-frequency class, resulting in potential misclassification of positive and negative features within this class. Consequently, our model does not exhibit assured superiority over existing models in terms of performance on high-frequency classes. However, when the low-frequency classes hold significant importance in the given task, our model proves to be more suitable.

## 4. Conclusions

Automated segmentation plays a crucial role in clinical settings as it facilitates precise delineation of internal tissue structures and lesions, providing valuable assistance to physicians in accurate diagnosis. In the present study, we propose an end-to-end network model, namely WA-ResUNet, for performing multi-class semantic segmentation on uterine magnetic resonance imaging (MRI) datasets. Enhancing the learning rate to effectively capture salient features, while mitigating the adverse impact of irrelevant features, results in an increased mean Intersection over Union (mIoU) and notably advances the performance in regard to the low-frequency class. Additionally, although there may be a decrease in the efficacy for the high-frequency class, this model proves to be well-suited for medical tasks where the significance of the low-frequency class cannot be undermined.

Furthermore, we introduce a novel adaptive class weight formula, based on the number of images contained in the class, to address the issue of imbalanced distribution in multi-class semantic segmentation. This approach aims to enhance the overall performance of semantic segmentation without compromising sensitivity or precision. In our dataset, the validity and generalization of the formula were verified. The impact of the formula on segmentation performance was evident when comparing its use with the absence of its application. The formula effectively enhances the segmentation performance of the low-frequency class, while not necessarily diminishing the performance of the high-frequency class. Furthermore, the experimental results demonstrated that this approach is compatible with the majority of existing models, making it a versatile and applicable method.

In the task undertaken in this experimental study, a notable advance of over 20% in Intersection over Union (IoU), specifically for nabothian cysts, was observed through the integration of the class weight formula and the WA-ResUNet model, resulting in an improved mean Intersection over Union (mIoU). While we achieved our initial objective, our aim is to further enhance the efficacy of the model and address the challenges associated with the long-tailed distribution. Consequently, our future endeavors will focus on rectifying the issue of high-frequency class loss stemming from the model’s excessive emphasis on low-frequency classes within the WA-ResUNet framework. Evidently, the integration of the weight formula and attention mechanisms in the WA-ResUNet enables notable improvements in the performance of the low-frequency class; thus, effectively addressing the challenge of class imbalance. However, as a consequence of increased attention and weight assigned to the low-frequency class, there may be a certain loss of attention towards the high-frequency classes. Consequently, this loss of attention may result in a reduction in IoU for the high-frequency class. Therefore, our future research aims to achieve a better balance between the weight relationship and attention levels of both the high-frequency and low-frequency classes. For instance, dynamic adjustment of the weights could be achieved by modifying the dynamic coefficient, based on the error from the previous iteration during the training process. Furthermore, we aim to extend the application of WA-ResUNet to address other semantic segmentation tasks characterized by long-tailed distribution issues, as this is a prevalent challenge in the field of medicine. Concurrently, we encourage fellow researchers to explore the utilization of WA-ResUNet in diverse tasks and leverage our weight formula to enhance the performance of low-frequency classes, as this approach holds promising implications for numerous datasets.

## Figures and Tables

**Figure 1 bioengineering-10-00945-f001:**
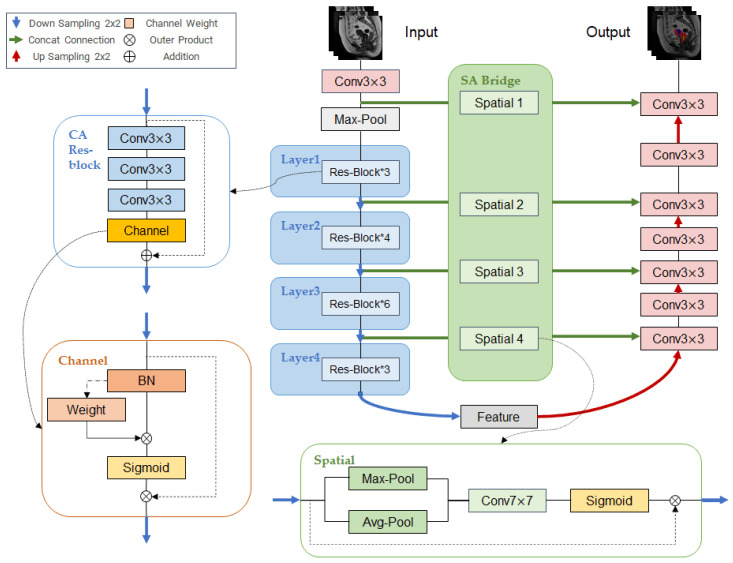
WA-ResUNet Network Structure.

**Figure 2 bioengineering-10-00945-f002:**
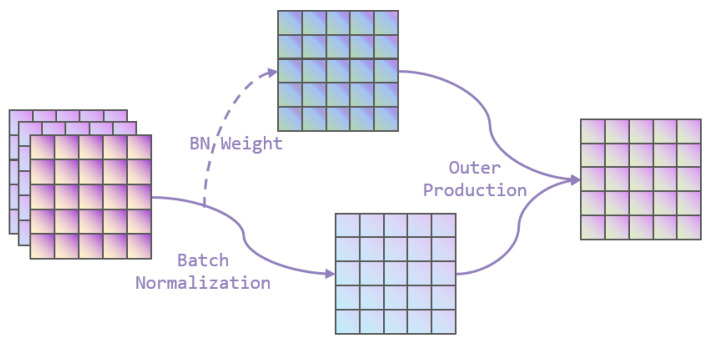
Channel Attention Module.

**Figure 3 bioengineering-10-00945-f003:**
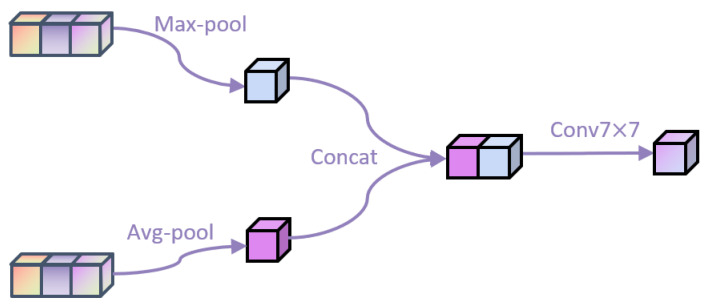
Spatial Attention Module.

**Figure 4 bioengineering-10-00945-f004:**
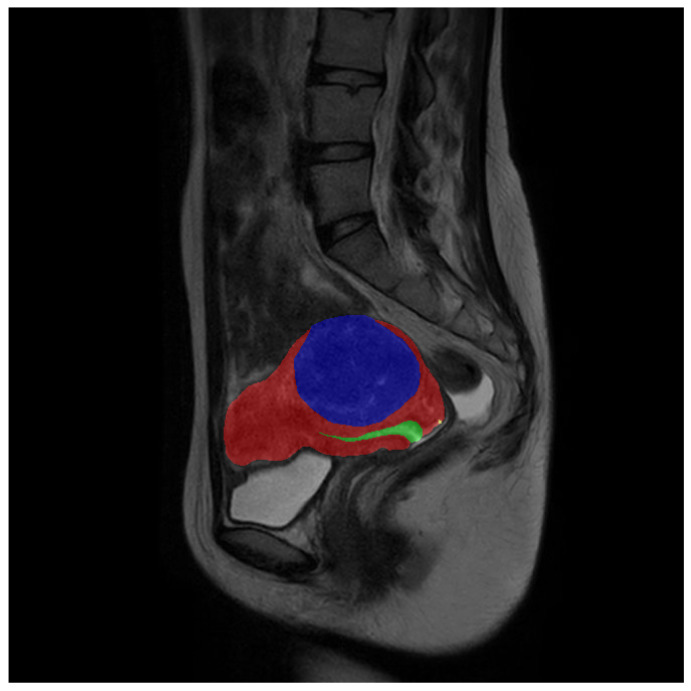
An example of our dataset.

**Figure 5 bioengineering-10-00945-f005:**
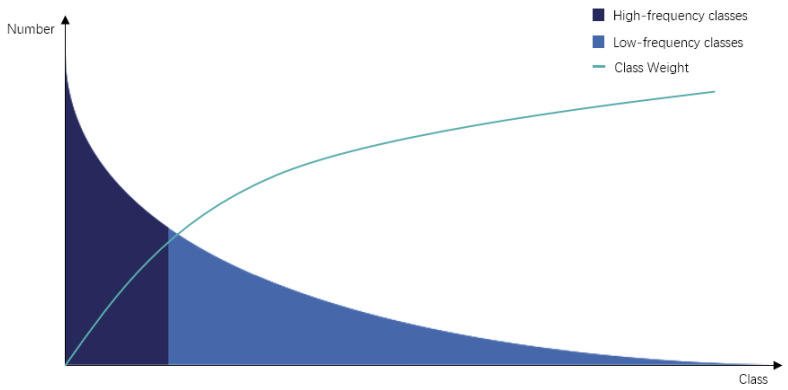
The relationship between long-tailed distribution and class weight.

**Figure 6 bioengineering-10-00945-f006:**
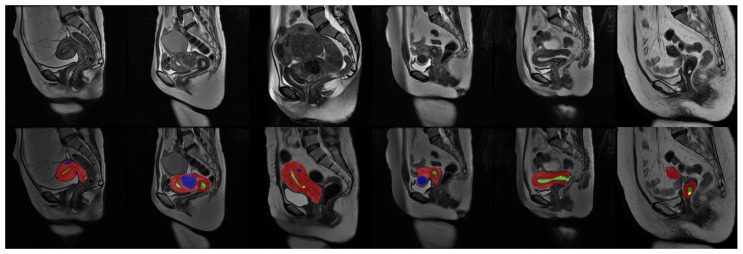
Data sample: The first line is the original image, and the second line is the real value image after visualization. During visualization, the uterine wall was covered in red, the uterine cavity was covered in green, myomas of the uterus were covered in blue, and nabothian cysts were covered in yellow.

**Figure 7 bioengineering-10-00945-f007:**
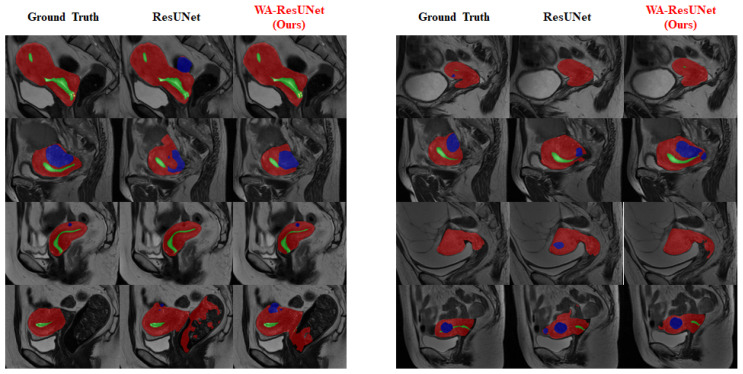
Comparison of visual inference results—overall: The three columns of the images show the predicted values of the expected baseline and the predicted values of our proposed model, respectively. During visualization, the uterine wall was covered in red, the uterine cavity was covered in green, myomas of the uterus were covered in blue, and nabothian cysts were covered in yellow.

**Figure 8 bioengineering-10-00945-f008:**
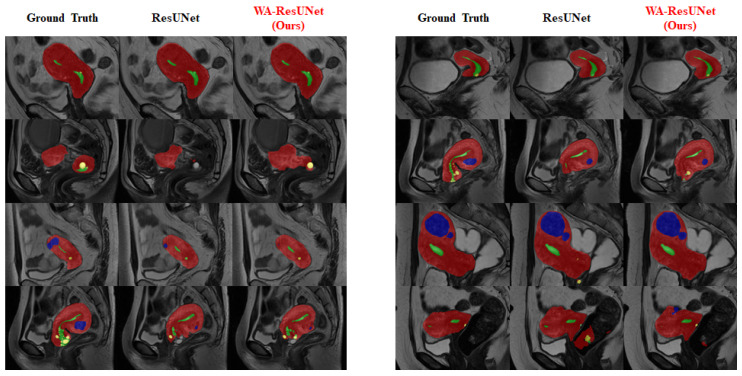
Comparison of visual inference results—nabothian cyst: The three columns of the images show the predicted values of the expected baseline and the predicted values of our proposed model, respectively. During visualization, the uterine wall was covered in red, the uterine cavity was covered in green, myomas of the uterus were covered in blue, and nabothian cysts were covered in yellow.

**Table 1 bioengineering-10-00945-t001:** Comparison of the number of pixels and the number of images in different classes of uterine MRI dataset and the related weight.

	Uterine Wall	Uterine Cavity	Myomas of Uterus	Nabothian Cysts
Number of pixels	13,501,701	659,031	10,655,977	39,608
Weight based on pixels	0.0027	0.0564	0.0034	0.9375
Number of images	1603	869	1090	201
Weight based on images	0.081	0.150	0.120	0.649

**Table 2 bioengineering-10-00945-t002:** The number of pixels in the example image.

	Uterine Wall	Uterine Cavity	Myomas of Uterus	Nabothian Cysts
Number of pixels	11,202	743	10,634	10

**Table 3 bioengineering-10-00945-t003:** Results of WA-ResUNet ablation experiment Overall results-IoU (The result in parentheses is the change from the previous row).

U-Net [6]	ResUNet [9]	Weight Formula	Channel Attention	Spatial Attention	Nabothian Cysts	Overall
✓					10.57	54.69
✓	✓				8.09 (−2.48)	56.06 (+1.37)
✓	✓	✓			12.98 (+4.89)	56.31 (+0.25)
✓	✓	✓	✓		23.80 (+10.82)	60.13 (+3.82)
✓	✓	✓	✓	✓	29.96 (+6.16)	61.23 (+1.10)

**Table 4 bioengineering-10-00945-t004:** Segmentation results of ResUNet, ResUNet+ Channel Attention, ResUNet+ Spatial Attention, and WA-ResUNet (ResUNet+ Both of them) when the weight of all classes is 1 (The bold value is the optimal value).

		Uterine Wall	Uterine Cavity	Myomas of Uterus	Nabothian Cysts	Overall
IoU	No Attention	62.78	48.76	61.13	8.09	45.19
	Channel Attention	62.26	**48.96**	60.15	15.63	46.75
	Spatial Attention	63.40	45.25	**62.07**	14.37	46.27
	Both of them	**63.65**	44.17	59.50	**20.25**	**46.89**
Sensitivity	No Attention	81.05	53.58	81.45	17.04	58.28
	Channel Attention	82.1	**54.89**	80.30	19.41	59.18
	Spatial Attention	80.80	49.32	**82.89**	23.77	**59.20**
	Both of them	**82.61**	48.50	78.70	**25.84**	58.91
Precision	No Attention	73.59	84.40	71.03	13.02	60.51
	Channel Attention	71.03	81.08	69.74	51.36	68.30
	Spatial Attention	**74.65**	**84.59**	**71.18**	26.66	64.27
	Both of them	73.50	83.19	70.98	**52.95**	**70.16**

**Table 5 bioengineering-10-00945-t005:** Segmentation results of ResUNet, ResUNet+ Channel Attention, ResUNet+ Spatial Attention, and WA-ResUNet (ResUNet+ Both of them) in new weight-IoU (The bold value is the optimal value).

		Uterine Wall	Uterine Cavity	Myomas of Uterus	Nabothian Cysts	Overall
IoU	No Attention	61.71	47.02	60.32	12.98	45.51
	Channel Attention	**62.97**	50.56	**63.80**	23.80	50.28
	Spatial Attention	62.09	48.12	60.27	16.33	46.70
	Both of them	62.22	**52.82**	61.66	**29.96**	**51.67**
Sensitivity	No Attention	82.37	51.11	79.48	14.28	56.81
	Channel Attention	**83.63**	55.00	**80.50**	26.34	61.37
	Spatial Attention	81.78	52.28	79.61	18.94	58.15
	Both of them	83.23	**57.59**	79.27	**33.70**	**63.45**
Precision	No Attention	71.11	85.28	71.46	62.85	72.68
	Channel Attention	71.82	86.21	**75.48**	73.06	**76.64**
	Spatial Attention	**72.05**	85.78	71.31	59.32	72.12
	Both of them	71.14	**86.44**	73.53	**74.05**	76.29

**Table 6 bioengineering-10-00945-t006:** Comparison of segmentation results between WA-ResUNet and other models in new weight-IoU (The bold value is the optimal value. Underscores indicate the next best data.).

	Uterine Wall	Uterine Cavity	Myomas of Uterus	Nabothian Cysts	Overall
Baseline	61.71	47.02	60.32	12.98	45.51
Channel-UNet [41]	**65.68**	48.42	64.12	22.74	50.24
Ce-net [11]	63.2	48.59	61.47	14.94	47.05
UNet++ [12]	64.95	**60.69**	63.56	0.21	47.35
MultiResUNet [14]	63.35	51.54	61.48	26.02	50.60
Resunet++ [13]	58.28	51.22	55.05	16.63	45.30
DC-UNet [15]	65.13	52.47	**66.84**	21.14	51.40
WA-ResUNet	62.22	52.82	61.66	**29.96**	51.67

**Table 7 bioengineering-10-00945-t007:** Comparison of segmentation results between WA-ResUNet and other attention mechanisms in new weight-IoU (The bold value is the optimal value. Underscores indicate the next best data.).

	Uterine Wall	Uterine Cavity	Myomas of Uterus	Nabothian Cysts	Overall
Baseline	61.71	47.02	60.32	12.98	45.51
Cbam Att [18]	60.41	40.08	56.86	12.01	42.34
Aff Att [20]	59.14	42.62	57.67	14.82	43.56
Coo Att [21]	61.24	52.82	61.93	13.48	47.37
PSA-S [22]	62.24	50.13	60.91	4.38	44.42
CSAB Att [25]	62.81	50.84	59.34	20.34	48.33
CSAB+Nam [24,25]	**64.23**	46.57	62.77	17.6	47.79
SimAM+Nam [23,24]	62.88	52.25	63.28	20.66	49.77
SAB+Nam [24,25]	63.37	52.38	**63.5**	18.4	49.41
Coo+Nam [21,24]	61.51	**57.74**	56.82	12.15	47.06
WA-ResUNet	62.22	52.82	61.66	**29.96**	**51.67**

**Table 8 bioengineering-10-00945-t008:** Segmentation results of ResUNet and WA-ResUNet in different weights (The bold value is the optimal value).

			Uterine Wall	Uterine Cavity	Myomas of Uterus	Nabothian Cysts	Overall
IoU	ResUNet [9]	Weight 1	**62.78**	**48.76**	**61.13**	8.09	45.19
		Weight 2	27.65	34.19	37.33	1.42	25.15
		Weight 3	61.71	47.02	60.32	**12.98**	**45.51**
	WA-ResUNet	Weight 1	**63.65**	44.17	59.50	20.25	46.89
		Weight 2	33.65	33.09	42.59	7.27	29.15
		Weight 3	62.22	**52.82**	**61.66**	**29.96**	**51.67**
Sensitivity	ResUNet [9]	Weight 1	81.05	53.58	**81.45**	17.04	58.28
		Weight 2	**87.94**	**55.90**	76.56	**77.66**	**74.52**
		Weight 3	82.37	51.11	79.48	14.28	56.81
	WA-ResUNet	Weight 1	82.61	48.50	78.70	25.84	58.91
		Weight 2	**88.84**	**60.02**	70.20	**67.61**	**71.67**
		Weight 3	83.23	57.59	**79.27**	33.70	63.45
Precision	ResUNet [9]	Weight 1	**73.59**	84.40	71.03	13.02	60.51
		Weight 2	28.74	46.81	42.15	1.43	29.78
		Weight 3	71.11	**85.28**	**71.46**	**62.85**	**72.68**
	WA-ResUNet	Weight 1	**73.50**	83.19	70.98	52.95	70.16
		Weight 2	35.31	42.92	51.23	9.47	34.73
		Weight 3	71.14	**86.44**	**73.53**	**74.05**	**76.29**

**Table 9 bioengineering-10-00945-t009:** Comparison of segmentation results of representative models in default weight and new weight-IoU.

		Uterine Wall	Uterine Cavity	Myomas of Uterus	Nabothian Cysts	Overall
Channel-UNet [41]	Default weight	65.99	54.55	61.21	15.32	49.27
	Our weight	65.68 (−0.31)	48.42 (−6.13)	64.12 (+2.91)	22.74 (+7.42)	50.24 (+0.97)
MultiResUNet [14]	Default weight	62.24	39.69	60.06	4.18	41.54
	Our weight	63.35 (+1.11)	51.54 (+11.85)	61.48 (+1.42)	26.02 (+21.84)	50.60 (+9.06)
UNet++ [12]	Default weight	66.81	53.12	64.52	0.0	46.11
	Our weight	64.95 (−1.86)	60.69 (+7.57)	63.56 (-0.96)	0.21 (+0.21)	47.35 (+1.24)
Ce-net [11]	Default weight	62.98	45.99	56.05	11.67	44.17
	Our weight	63.2 (+0.22)	48.59 (+2.6)	61.47 (+5.42)	14.94 (+3.27)	47.05 (+2.88)
Resunet++ [13]	Default weight	57.56	43.59	57.31	13.84	43.08
	Our weight	58.28 (+0.72)	51.22 (+7.63)	55.05 (−2.26)	16.63 (+2.79)	45.30 (+2.22)
DC-UNet [15]	Default weight	65.76	50.83	62.7	15.0	48.57
	Our weight	65.13 (−0.63)	52.47 (+1.64)	66.84 (+4.14)	21.14 (+6.14)	51.40 (+2.82)

**Table 10 bioengineering-10-00945-t010:** Segmentation results of U-Net, ResUNet and WA-ResUNet (The bold value is the optimal value).

		Uterine Wall	Uterine Cavity	Myomas of Uterus	Nabothian Cysts	Overall
IoU	U-Net [6]	60.66	41.91	60.83	10.57	43.49
	ResUNet [9]	**62.78**	48.76	61.13	8.09	45.19
	WA-ResUNet	62.22	**52.82**	**61.66**	**29.96**	**51.67**
Sensitivity	U-Net [6]	79.54	45.51	76.34	18.89	55.07
	ResUNet [9]	81.05	53.58	**81.45**	17.04	58.28
	WA-ResUNet	**83.23**	**57.59**	79.27	**33.70**	**63.45**
Precision	U-Net [6]	71.88	84.12	72.96	19.35	62.08
	ResUNet [9]	**73.59**	84.40	71.03	13.02	60.51
	WA-ResUNet	71.14	**86.44**	**73.53**	**74.05**	**76.29**

**Table 11 bioengineering-10-00945-t011:** Results of comparison between WA-ResUNet model and other models -IoU and FPS (The bold value is the optimal value.).

	Uterine Wall	Uterine Cavity	Myomas of Uterus	Nabothian Cysts	Overall	FPS
Channel-UNet [41]	65.99	**54.55**	61.21	15.32	49.27	4.12
Ce-net [11]	62.98	45.99	56.05	11.67	44.17	26.95
UNet++ [12]	**66.81**	53.12	**64.52**	0.0	46.11	4.08
MultiResUNet [14]	62.24	39.69	60.06	4.18	41.54	6.49
Resunet++ [13]	57.56	43.59	57.31	13.84	43.08	12.16
DC-UNet [15]	65.76	50.83	62.7	15.0	48.57	5.98
WA-ResUNet	62.22	52.82	61.66	**29.96**	**51.67**	6.04

**Table 12 bioengineering-10-00945-t012:** Comparison of the number of pixels and the number of images in different classes of the abdominal organs dataset.

	Liver	Left Kidney	Right Kidney	Spleen
Number of pixels	1,237,685	133,279	139,442	183,227
Number of images	345	213	222	194

**Table 13 bioengineering-10-00945-t013:** Segmentation results of ResUNet and WA-ResUNet on abdominal organs dataset (The bold value is the optimal value).

		Liver	Left Kidney	Right Kidney	Spleen	Overall
IoU	ResUNet [9]	**85.48**	74.78	78.02	**72.39**	77.67
	WA-ResUNet	82.73	**86.96**	**81.03**	71.17	**80.47**
Sensitivity	ResUNet [9]	**94.38**	77.2	**93.12**	**80.62**	86.33
	WA-ResUNet	86.16	**90.25**	91.38	78.76	**86.64**
Precision	ResUNet [9]	90.06	95.97	82.79	87.63	89.11
	WA-ResUNet	**95.4**	**95.98**	**87.74**	**88.07**	**91.80**

## Data Availability

It is not announced for the time being, because there are further research plans. If necessary, you can contact the author to obtain the data.

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
