# Peer review of "WA-ResUNet: A Focused Tail Class MRI Medical Image Segmentation Algorithm"

_bioengineering, 2023, doi:10.3390/bioengineering10080945_

Round 1
Reviewer 1 Report
I appreciate your work. The problem you address is very important.
Figure 1 is useless. You can either remove it or change it. It also contain a threshold that is not defined in terms of values or percentage.
Author Response
We would like to thank you for your careful reading, helpful comments, and constructive suggestions, which has significantly improved the presentation of our manuscript. And we also thank the reviewer for reading our paper carefully and giving the above positive comments. Regarding your suggestion, we have the following reply:
Point 1. Figure 1 is useless. You can either remove it or change it. It also contain a threshold that is not defined in terms of values or percentage.
Response 1: We have removed it now according to the reviewer’s suggestion. Thanks for your suggestion and apologies for our unprofessional presentation.

Reviewer 2 Report
The work is well described and very deepened from the scientific point of view. Many comparisons with the existing models in literature have been performed and this gives a remark to the whole work.
I have only some minor comments:
- Abstract (line 11): I think the ',' has to be sobstituted with '.'
-Abstarct: please explain IoU
- All the acronyms have to be explicited when first used (i.e. line 44 FCN, line 22 HPV, line 90 IoU etc) .
-Table 2 is not clear to me. What do the values reported stand for? Please explain better
- In the Results and Discussion section, please explicit the limitation and future directions.
The English language must be improved.
Author Response
We would like to thank you for your careful reading, helpful comments, and constructive suggestions, which has significantly improved the presentation of our manuscript. And we also thank the reviewer for reading our paper carefully and giving the above positive comments. Regarding your suggestion, we have the following reply:
Point 1. Abstract (line 11): I think the ',' has to be sobstituted with '.'
Response 1: We have changed it now according to the reviewer’s suggestion. Thanks for your suggestion and apologies for our mistake.
Point 2.&3. Abstarct: please explain IoU
All the acronyms have to be explicited when first used (i.e. line 44 FCN, line 22 HPV, line 90 IoU etc) .
Response 2&3: Thank you very much for your advice. We have supplemented the full name of the abbreviation where it first appeared.(line 1, line 12, line 22, line27, line 46, line 54, line 59, line 94 and line 449)
Point 3. Table 2 is not clear to me. What do the values reported stand for? Please explain better
Response 3: We feel sorry for the confusion brought to the reviewer.
Actually,the first line represents the result of the U-Net model, the second line represents the result of using Res-Block on the basis of the U-Net model, the third line represents the result of using Res-Block and the weight formula on U-Net simultaneously. The fourth line represents the addition of CA Res-Block and weight formulas with channel attention mechanism on the basis of U-Net. The fifth line represents the complete model, that is, using CA Res-Block, weight formula, and SA Bridge simultaneously on U-Net.
We've added it to Section 3.2 (line 322-line 327)
Point 4. In the Results and Discussion section, please explicit the limitation and future directions.
Response 4: Thank you for your suggestion.
Overall, for limitation, we reckon that WA-ResUNet can better compensate for the loss caused by long-tailed distribution. By improving the learning rate of the network for effective features while suppressing the negative impact from invalid features, the learning ability of the network on low-frequency classes can be improved, and the network is able to obtain significant improvement on low-frequency classes though there is a slight decrease in performance on high-frequency classes. We consider that this is due to the reduction of the weight of the high-frequency class, which leads to the positive and negative features in the high-frequency class can be distinguished incorrectly. This means that our model will not necessarily perform better on high-frequency classes compared to existing models, but when low-frequency classes are important enough in the task, our model will be more appropriate.
We've added it to the end of Section 3(line 502-line 512). In addition, we also added a more detailed discussion of the limitations of the model to the analysis of the experimental data.(line 364-line 367, line 376-line 378 and line 410-line 412)
For future limitation, we mainly introduce our ideas and plans for the next step in the conclusion section. (line 543-line 551)
Obviously, WA-ResUNet improves the low-frequency class under the influence of weight formula and attention mechanisms, which means that the model can better overcome the problem of class imbalance. At the same time, due to the increase of attention and weight of the low-frequency class, the high-frequency classes gain a certain loss of attention. This may lead to a decrease in the IoU of the high-frequency class, so next we hope to better balance the weight relationship between high frequency and low frequency and the degree of attention. For example, the weight is dynamically changed by changing the dynamic coefficient through the error of the previous round during the training process. In addition, we hope to use WA-ResUNet to solve other semantic segmentation tasks with long-tailed distribution problems because it is a common problem in medicine. At the same time, we also welcome other researchers to apply WA-ResUNet to more tasks and apply our weight formula to improve the performance of the low-frequency class, which makes sense in many datasets.

Reviewer 3 Report
I am really grateful to review this manuscript. In my opinion, this manuscript can be published once some revisions are done successfully. I made two suggestions and I would like to ask your kind understanding. This study used 7234 magnetic resonance images, introduced a Resnet-based Unet with an attention mechanism weighted for class distribution, and achieved the intersection over union (IOU) of 51.7% for the segmentation of four uterine components (cavity, myomas, nabothian cysts and wall) (Table 9). I would argue that this is a rare achievement. However, firstly, the performance of this model was better than those of existing models only for one particular class (nabothian cysts), which would put a significant restriction on the application of this model. In this context, I would like to ask the author to address this issue in greater detail in Discussion. Secondly, frames per second (FPS) is gaining increasing attention as another important metric together with the IOU but model comparison in this metric is missing. In this vein, I would like to ask the authors to add the table of model comparison in terms of FPS in the manuscript.
Minor editing of English language required
Author Response
We would like to thank you for your careful reading, helpful comments, and constructive suggestions, which has significantly improved the presentation of our manuscript. And we also thank the reviewer for reading our paper carefully and giving the above positive comments. Regarding your suggestion, we have the following reply:
Point 1. firstly, the performance of this model was better than those of existing models only for one particular class (nabothian cysts), which would put a significant restriction on the application of this model. In this context, I would like to ask the author to address this issue in greater detail in Discussion.
Response 1: Thank you very much for your advice.
Overall, for limitation, we reckon that WA-ResUNet can better compensate for the loss caused by long-tailed distribution. By improving the learning rate of the network for effective features while suppressing the negative impact from invalid features, the learning ability of the network on low-frequency classes can be improved, and the network is able to obtain significant improvement on low-frequency classes though there is a slight decrease in performance on high-frequency classes. We consider that this is due to the reduction of the weight of the high-frequency class, which leads to the positive and negative features in the high-frequency class can be distinguished incorrectly. This means that our model will not necessarily perform better on high-frequency classes compared to existing models, but when low-frequency classes are important enough in the task, our model will be more appropriate.
We've added it to the end of Section 3(line 502-line 512). In addition, we also added a more detailed discussion of the limitations of the model to the analysis of the experimental data.(line 364-line 367, line 376-line 378 and line 410-line 412)
Point 2. I would like to ask the authors to add the table of model comparison in terms of FPS in the manuscript.
Response 2: We appreciate your constructive suggestions. We have added the indicator FPS in Table 11 of the comparison experiment and briefly explained it below. (line 459-line 465)

Reviewer 4 Report
This paper introduces a weighted attention ResUNet (WA-ResUNet), for the segmentation of greyscale MRI images. The main motivation and contribution towards introducting class weights (according to the number of images in the class), is the imbalanced (long-tailed) distribution of class data in practice. As such, assigning higher weights to rarer classes should increase the amount of attention paid to such classes. In a uterine MRI dataset of 3206 MRI images, WA-ResUNet improved IoU on the rarest class (Nabothian cysts) by close to 22%, and overall IoU by over 6.5%. Additional experiments on a public abdominal organ dataset were also presented.
While the various extensions to the basic ResUNet were shown to aid classification of the rarest class (Table 2), some issues might be considered:
1. The key technical novelties claimed appear to be the weight formula, channel attention and spatial attention. However, all these appear to have been applied in the medical image segmentation domain in some form. To begin with, weighing a class by the inverse ratio of instances is quite common in deep learning classification tasks. As for channel/spatial attention, a number of past works have explored it with U-Nets, e.g.
Zhao, P., Zhang, J., Fang, W., & Deng, S. (2020). SCAU-net: spatial-channel attention U-net for gland segmentation. Frontiers in Bioengineering and Biotechnology, 8, 670.
Guo, C., Szemenyei, M., Yi, Y., Wang, W., Chen, B., & Fan, C. (2021, January). Sa-unet: Spatial attention u-net for retinal vessel segmentation. In 2020 25th international conference on pattern recognition (ICPR) (pp. 1236-1242). IEEE.
etc.
As such, the specific contributions of this work might be clarified in greater detail.
2. In Section 2.4, the class weighing is by number of images containing a class, and not the number of pixels of that class. This might be clarified since it appears possible that some particular class might be rare, but nevertheless occurs in the majority or all the images, just with very few pixels per image. However, in such a case, its weight would remain relatively low and similar to the more-common classes.
3. Related to the above, it might be considered to explore how the IoU of the various classes varies, as the weight formula applied to each class changes (both increasing and decreasing some multiplier to the inverse ratio)
4. In Section 3.2, the ablation experiment purports to show the contribution of the three extensions (weight formula, channel attention, spatial attention) on IoU performance of the rarest class (Nabothian cysts). However, it is not clear if the extensions help the performance on the remaining classes. If possible, the chance in performance for all classes might be presented (Table 3 shows partial results, but without ablating the weight formula).
5. Table 4 suggests that the proposed WA-ResUNet improves performance significantly only for the rarest class, and existing models perform better on the other classes. Moreover, although the overall IoU is supposedly superior to the comparison models, this seems to be due to all classes being weighted equally in calculation overall IoU. It is unclear whether this is desirable, and might be clarified.
N/A
Author Response
We would like to thank you for your careful reading, helpful comments, and constructive suggestions, which has significantly improved the presentation of our manuscript. And we also thank the reviewer for reading our paper carefully and giving the above positive comments. Regarding your suggestion, we have the following reply:
Point 1&2&3. the specific contributions of this work might be clarified in greater detail.
In Section 2.4, the class weighing is by number of images containing a class, and not the number of pixels of that class. This might be clarified.
Related to the above, it might be considered to explore how the IoU of the various classes varies, as the weight formula applied to each class changes (both increasing and decreasing some multiplier to the inverse ratio)
Response 1&2&3: Thank you very much for your advice.
First of all, we apologize to you for not explaining the content of our article clearly. We believe that although the case-based class rebalancing method is widely used, the weight formula based on the number of images has not been proposed by researchers, and the theoretical basis of the two in semantic segmentation is different, we have introduced and explained it in more detail.
The method of assigning weights based on the number of instances is also not mutually independent in semantic segmentation, because two instances of the same graph may have some potential relations for example the location of them.
It has been added to line 265 to line 268.
At the same time, the shortcomings and problems of pixel-based weight formula in segmentation tasks are discussed in detail. To explain it more clearly, we added a new table(Table 2) and a new picture(Figure 4) and we added the data in Table 1. In summary, there are several reasons as followes:
- In medical imaging, the increase in a pixel does not necessarily add new features because the single pixel has no labels and model can not classify depending on one pixel.
- Conversely,we reckon thata small number of pixels is not directly related to a small number of features.
- In addition, in this dataset, pixel-level weight formula leads to too low weight of the model for high-frequency classes.
- Meanwhile, the weight formula based on the number of pixels does not conform to the characteristics of mutual independence.
Each point is explained and supplemented in more detail in Section 2.4. (line 233-line 274)
In addition, regarding the problem raised by reviewers that there are fewer singleton pixels but more occurrences, we also added explanations in the paper.
Conversely,we reckon that a small number of pixels is not directly related to a small number of features, because if the number of singleton occurrences is large enough, the network can also learn enough positive information with a small number of singleton pixels, because, in this way, the common points between each singleton will be easier to find.
We have added it to line 239-line 247.
Point 4. In Section 3.2, it is not clear if the extensions help the performance on the remaining classes. If possible, the chance in performance for all classes might be presented (Table 3 shows partial results, but without ablating the weight formula).
Response 4: We appreciate your constructive suggestions. We have supplemented the data of relevant experiments and explained them. (line 336- line 346) We also add a new table (Table 4) for showing the result of experiment.
Point 5. Although the overall IoU is supposedly superior to the comparison models, this seems to be due to all classes being weighted equally in calculation overall IoU. It is unclear whether this is desirable, and might be clarified.
Response 5: We feel sorry for the confusion brought to the reviewer.
But we have some reasons for that.
- At present, most models use this method to calculate mIoU without weighting it.
- We reckon that in our experiment, of the four classes in our uterine MRI dataset, the lesion class is significantly more important than the organ class, because the main purpose of the task is to better help doctors diagnose the condition. Therefore, even if it is weighted, the weight of classes 4 and 3 should be increased.
- We think our modules are equally important. Especially the weight formula, in other networks, is able to improve the performance of some low frequency classes and some high frequency classes at the same time. For example, in DC-UNet and Ce-net, the splitting effect of class 3 and class 4 has been improved together.This means our model will still have a better overall effect even if the weight is added.
We've added it to Section 3.2 (line 413-line 419).

Round 2
Reviewer 3 Report
I am really grateful to review this manuscript. In my opinion, this manuscript can be published in current form.
Minor editing of English language required
Author Response
We would like to thank you for your careful reading, helpful comments, and constructive suggestions.
Regarding your suggestion, we have revised our manuscript.
Thank you for your recognition of my work.

Reviewer 4 Report
We thank the authors for addressing our previous comments.
Author Response
We would like to thank you for your careful reading, helpful comments, and constructive suggestions.
Thank you for your recognition of my work.
